# Novel Histone-Based DNA Carrier Targeting Cancer-Associated Fibroblasts

**DOI:** 10.3390/polym12081695

**Published:** 2020-07-29

**Authors:** Alexey Kuzmich, Olga Rakitina, Dmitry Didych, Victor Potapov, Marina Zinovyeva, Irina Alekseenko, Eugene Sverdlov

**Affiliations:** 1Institute of Molecular Genetics, Russian Academy of Sciences; 2, Kurchatov Square, 123182 Moscow, Russia; irina.alekseenko@mail.ru (I.A.); edsverd@gmail.com (E.S.); 2Shemyakin-Ovchinnikov Institute of Bioorganic Chemistry, Russian Academy of Sciences; 16/10, Miklukho-Maklaya, 117997 Moscow, Russia; rakitinaolga97@gmail.com (O.R.); dmitrydid@gmail.com (D.D.); vk@ibch.ru (V.P.); mzinov@mail.ru (M.Z.); 3FSBI National Medical Research Center for Obstetrics, Gynecology and Perinatology named after Academician V.I. Kulakov Ministry of Healthcare of the Russian Federation, 117198 Moscow, Russia; 4National Research Center “Kurchatov Institute”, Akademika Kurchatova pl. 1, 123182 Moscow, Russia

**Keywords:** histone H2A, cancer-associated fibroblasts, gene therapy, targeted gene delivery, internalization, receptor-mediated endocytosis, beta-type platelet-derived growth factor receptor, transfection, targeting peptide, non-viral gene delivery

## Abstract

Nuclear proteins, like histone H2A, are promising non-viral carriers for gene delivery since they are biocompatible, biodegradable, bear intrinsic nuclear localization signal, and are easy to modify. The addition of surface-protein-binding ligand to histone H2A may increase its DNA delivery efficiency. Tumor microenvironment (TME) is a promising target for gene therapy since its surface protein repertoire is more stable than that of cancer cells. Cancer-associated fibroblasts (CAFs) are important components of TME, and one of their surface markers is beta-type platelet-derived growth factor receptor (PDGFRβ). In this study, we fused histone H2A with PDGFRβ-binding peptide, YG2, to create a novel non-viral fibroblast-targeting DNA carrier, H2A-YG2. The transfection efficiency of histone complexes with pDNA encoding a bicistronic reporter (enhanced green fluorescent protein, EGFP, and firefly luciferase) in PDGFRβ-positive and PDGFRβ-negative cells was estimated by luciferase assay and flow cytometry. The luciferase activity, percentage of transfected cells, and overall EGFP fluorescence were increased due to histone modification with YG2 only in PDGFRβ-positive cells. We also estimated the internalization efficiency of DNA-carrier complexes using tetramethyl-rhodamine-labeled pDNA. The ligand fusion increased DNA internalization only in the PDGFRβ-positive cells. In conclusion, we demonstrated that the H2A-YG2 carrier targeted gene delivery to PDGFRβ-positive tumor stromal cells.

## 1. Introduction

Gene therapy, which is based on the delivery of nucleic acids to target cells in the patient’s body, is a promising approach for cancer treatment. Currently, viral delivery systems are most commonly used in gene therapy systems. They provide an efficient means of transport of genetic material into the target cells owing to their ability to transfer their genome into the cells of the host. However, viral carriers have several disadvantages, including cytotoxicity, potential carcinogenicity, difficulties in production and storage, as well as immunogenicity-related problems associated with the re-administration of such drugs [1,2]. In this regard, non-viral delivery systems, based on cationic polymers, cationic lipids or peptides, and nuclear DNA-binding proteins, such as histones, are promising alternatives. Nuclear proteins have a number of additional advantages, including biocompatibility, biodegradability, ability to deliver the cargo specifically to the nucleus due to intrinsic nuclear localization signal (NLS), and, finally, ease of desired sequence changes. 

Positively charged histones bind nonspecifically to DNA and neutralize its negative charge, thus allowing the DNA to enter cells through a negatively charged cell membrane. Subsequently, their NLS facilitates the entry of the DNA into the nucleus [3]. We have previously confirmed the ability of the human histone H2A to bind to pDNA [4] and transfect eukaryotic cells [5].

The efficiency and specificity of DNA delivery into certain cells may be increased using ligands that selectively bind to receptors exposed on the cell surface and induce internalization of these receptors [6,7]. In tumors, the cancer cells are usually considered the main target of gene therapy. The first cancer gene therapy drugs approved for clinical use (Gendicine, Oncorine, and Imlygic) were administered using intratumoral injection and were directed at cancer cells. However, it is also possible to use gene delivery targeted at tumor stromal cells, such as cancer-associated fibroblasts (CAFs) [8]. Stromal cells modified through gene delivery can serve as a long-term source of the antitumor compounds inside the tumor. In addition, the delivery efficiency of stroma-targeted therapy may be higher than that of delivery into cancer cells owing to the stability of stromal cell markers. Moreover, the stroma-targeted carriers would be more universal than the cancer cell-targeted carriers and, therefore, applicable to various types of tumors. At present, the number of studies on the use of CAF-targeted nucleic acid delivery is limited. In a previous study [9], lipid-coated protamine complexes loaded with pDNA encoding the secreted form of the tumor necrosis factor-related apoptosis-inducing ligand (sTRAIL) were used. Anisamide was conjugated on the surface of the particles as a ligand for the sigma-receptor, which is overexpressed on both CAFs and cancer cells. The study demonstrated the effectiveness of the system in a murine tumor model. As a result of the triple administration of the complexes, more than half of the CAFs expressed the therapeutic gene and produced cytokine, which, in turn, induced apoptosis of neighboring tumor cells, but not the CAFs. Moreover, residual CAFs were reverted to a non-activated state by sTRAIL. Another example of CAF-targeted nucleic acid delivery is a recent work [10], where the authors used peptide nanoparticles bearing CXCL12-targeted siRNA. These nanoparticles incorporated monoclonal anti-fibroblast activation protein α (FAPα) antibodies and were targeted specifically toward CAFs because they overexpress the FAPα receptor. The authors demonstrated that the absorption of the complexes was FAP-dependent and that tumor angiogenesis, as well as tumor cell invasion and migration, was significantly inhibited in vitro. In addition, these effects led to the suppression of metastasis in vivo. Thus, CAF-targeted gene delivery represents a promising approach to cancer therapy.

The β-type platelet-derived growth factor receptor, PDGFRβ, is one of the well-studied CAF markers [11]. A wide variety of ligands are known to bind to PDGFRβ, many of which are linear [12,13] and cyclic [14] peptides or antibody mimetic (affibody) molecules [11]. These ligands have also been used to deliver cytotoxic proteins and radioisotopes to the tumor stroma. However, they have not been used as a part of a gene delivery system. The linear peptide YG2 (YIPLPPPRRPFFK) binds to PDGFRβ-positive cells and accumulates inside them [13]. We hypothesized that the fusion of histone H2A with YG2 increases the efficiency of DNA delivery into CAFs. To test this hypothesis, we generated histone H2A fused with YG2 and analyzed the effect of the YG2 peptide on the efficiency of histone H2A-mediated DNA delivery into PDGFRβ-positive eukaryotic cells.

## 2. Materials and Methods

### 2.1. Materials

Plasmid pET30a(+) was obtained from Novagen (Madison, WI, USA), plasmid pGL3-BV was obtained from Promega (Madison, WI, USA), *Escherichia coli* DH5α strain was obtained from Invitrogen (Carlsbad, CA, USA), *E. coli* BL21(DE3) strain was obtained from Novagen. DMEM/F12, RPMI-1640, Fetal Bovine Serum (FBS), Trypan Blue Stain (0.4%), and Lipofectamine^®^ 2000 were obtained from Invitrogen. Antibiotic-Antimycotic (100×), Trypsin-EDTA, and Opti-MEM were obtained from Gibco^TM^ (Carlsbad, CA, USA); 5× passive lysis buffer (PLB) was obtained from Promega. All the oligonucleotide primers were synthesized by Evrogen (Moscow, Russia). All other chemical reagents were obtained commercially as reagent grade products. 

### 2.2. pDNA Production

Plasmid encoding human histone H2A (H2A-pET30) was previously obtained [5]. The gene encoding histone H2A fused with YG2 using the (Gly)_3_Ser linker and flanked by NdeI and HindIII restriction sites was generated with H2A-pET30 using a 2-step PCR and cloned into the pET30a(+) vector, resulting in the H2A-YG2-pET30 plasmid. PCR was performed using Encyclo polymerase (Evrogen) and H2A for and YG1 rev primers in the first step and H2A for and YG2 rev primers in the second step. The primer sequences are presented in Table 1. The fidelity of the H2A-YG2-encoding sequence was confirmed using Sanger sequencing.

The bicistronic plasmid pCMV-EGFP-P2A-luc2, encoding reporter proteins EGFP (enhanced green fluorescent protein) and firefly luciferase connected through a P2A peptide, was generated previously [15] based on the *EGFP* gene from pEGFP-N1 (Clontech Laboratories Inc. Mountain View, CA, USA), the *luc2* gene from pGL4.10 [luc2] (Promega), and P2A sequence [16]. 

All plasmids were transformed into *E. coli* DH5α strain for further manipulation. Plasmids H2A-pET30 and H2A-YG2-pET30 were extracted from bacterial cells using the Plasmid Miniprep kit (Evrogen) and then transformed into *E. coli* BL21(DE3) strain for protein synthesis. Plasmid pCMV_EGFP_P2A_luc2 was extracted from 200 mL of overnight culture using the EndoFree Plasmid Maxi kit (QIAgen, Venlo, The Netherlands). The DNA concentration was determined using a NanoDrop^TM^ 2000 (Thermo Scientific^TM^, Wilmington, DE, USA). The pGL3-BV plasmid was obtained from Promega.

### 2.3. Recombinant Histone Preparation

Bacterial cells containing H2A-pET30 or H2A-YG2-pET30 plasmid were propagated in the LB medium. Protein expression was induced by the addition of IPTG (isopropyl β-d-1-thiogalactopyranoside) to the final concentration of 1 mM. After 3 h of incubation at 37 °C, cells were harvested and lysed by sonication in lysis buffer (50 mM Tris-HCl, pH 7.5, 6 M urea, 1 mM EDTA, 200 mM NaCl, and 1 mM phenylmethanesulfonyl fluoride). The obtained lysate was centrifuged at 4000 rpm for 40 min to pellet the insoluble debris. To purify the clarified lysate from endotoxin, it was treated with HCl (final concentration of 0.25 M) for 1 h at 25 °C, centrifuged at 4000 rpm for 40 min to discard the insoluble pellet, and the pH of the solution was adjusted to 7.5 using 3 M Tris-base. Target proteins were purified in two steps using ion-exchange chromatography (SP Sepharose High-Performance resin (GE Healthcare, Danderyd, Sweden), 0.2–1 M NaCl gradient elution), followed by reversed-phase HPLC (Supelco BIO Wide Pore C18 column, Sigma-Aldrich, St. Louis, MO, USA, 5–80% acetonitrile gradient). H2A or H2A-YG2-containing fractions from the second chromatography were combined and dehydrated using a SpeedVac Concentrator (Savant Instruments Inc., Farmingdale, NY, USA). Subsequently, the proteins were dissolved in PBS buffer, and the protein concentration was quantified using the Protein Assay Dye Reagent Concentrate (Bio-Rad, Hercules, CA, USA) according to the manufacturer’s protocol. Protein purity was estimated electrophoretically.

### 2.4. Binding of H2A-YG2 to pDNA

H2A and H2A-YG2 were complexed with 1 μg of pGL3-BV plasmid at N/P (nitrogen to phosphate; charge) ratios of 0.5:1, 1:1, 2:1, 3:1, and 4:1 in PBS. The N/P ratios were calculated based on the number of arginine and lysine residues in the protein sequence and the number of phosphate groups in the DNA. The resulting complexes were incubated at 25 °C for 30 min, and then 1/5 of the resulting mixes were subjected to electrophoresis on a 0.8% agarose gel (1 × TAE). Next, the pDNA mobility was visualized using ethidium bromide staining.

### 2.5. Measurement of Size and Zeta Potential for Histone:pDNA Complexes

The complexes of histones H2A and H2A-YG2 with 20 μg of pCMV_EGFP_P2A_luc2 plasmid at N/P ratios of 2.5:1, 5:1, 10:1, and 15:1 were prepared in PBS (pH 7.4). Particle sizes of these complexes were measured by dynamic light scattering using a Brookhaven 90plus particle size analyzer (Brookhaven Instruments, Holtsville, NY, USA). Measurements were performed in a plastic cuvette at 25 °C in ten runs of 20 s duration each and analyzed in MSD (multimodal size distribution) analysis using 90Plus Particle sizing software. Zeta potential of the particles was measured by electrophoretic light scattering using a Brookhaven 90plus particle size analyzer with the BI-PALS module (Brookhaven Instruments, Holtsville, NY, USA) in the same conditions and calculated using PALS Zeta Potential Analyzer software (Ver, 5.78, Brookhaven Instruments, Holtsville, NY, USA).

### 2.6. Cell Culture

293T (human embryonic kidney transformed with SV40 T antigen, ATCC^®^ CRL-3216™), A375 (human melanoma, ATCC^®^ CRL-1619™), CT 26 (mouse colon carcinoma, ATCC^®^ CRL-2638™), and NIH/3T3 (mouse embryonic fibroblasts, ATCC^®^, CRL-1658™) cell lines were obtained from ATCC^®^ (Manassas, VA, USA).

293T, A375, and NIH/3T3 cell lines were cultured in DMEM/F12 medium (1:1) supplemented with 10% FBS and antibiotic-antimycotic solution in a 5% CO_2_ incubator at 37 °C under 95% humidity.

The CT 26 cell line was cultured in RPMI-1640 medium supplemented with 12.5% FBS and antibiotic-antimycotic solution in a 5% CO_2_ incubator at 37 °C under 95% humidity. 

All the cell lines were subcultured every 3–4 days using trypsin-EDTA. For mRNA content characterization and transfection experiments, cells were collected with trypsin-EDTA, stained with trypan blue stain (Invitrogen), and counted using the Countess II FL Automated Cell Counter (Invitrogen).

### 2.7. Analysis of PDGFRB mRNA Expression in Cell Lines Using RT-PCR

A total of 1 × 10^6^ cells were collected using trypsin-EDTA, homogenized using QIAshredder (QIAgen, Venlo, The Netherlands), and the total RNA was isolated using an RNeasy Plus Mini kit (QIAgen). The quality of the isolated RNA was evaluated by electrophoresis on a 1% agarose gel containing ethidium bromide. The RNA was reverse-transcribed with random decamer primers and Mint cDNA Synthesis kit (Evrogen). Real-time PCR was performed using qPCR mix-HS SYBR (Evrogen) according to the manufacturer’s instructions with a final primer concentration of 300 nM and 10 ng of the matrix. Primers Psmb2 for, Psmb2 rev, Psmb7 for, Psmb7 rev, Pdgfrb for, and Pdgfrb rev were used for murine cell lines, while primers HPRT for, HPRT rev, 18S for, 18S rev, PDGFRB for, and PDGFRB rev were used for human cell lines. PCR was performed using the LightCycler^®^ 96 real-time PCR cycler (Roche, Basel, Switzerland). All reactions were performed in triplicate according to the following protocol: heating at 95 °C for 5 min, followed by 45 cycles of denaturation at 95 °C for 10 s, annealing at 60 °C for 10 s, and extension at 72 °C for 10 s. Gene expression was quantified in N_0_ units using LinRegPCR software (Heart Failure Research Center, Amsterdam, The Netherlands), and the relative expression of *PDGFRB* (*Pdgfrb*) was calculated by dividing N_0_ for *PDGFRB* (*Pdgfrb*) by the geometric mean of N_0_ of housekeeping genes proteasome subunit beta type-2 and 7 (*Psmb2* and *Psmb7)*, respectively, for murine cell lines and hypoxanthine-guanine phosphoribosyltransferase (*HPRT*) and 18S ribosomal RNA (*18S* rRNA) for human cell lines. 

### 2.8. Transfection and Internalization Studies

To compare the internalization and transfection efficiency of H2A-YG2 and H2A, NIH/3T3 and 293T cell lines were transfected with the different carriers. The pCMV-EGFP-P2A-luc2 plasmid encoding EGFP and firefly luciferase was used for these experiments. 

The transfection efficiency was analyzed using luciferase activity assay and flow cytometry. To evaluate luciferase activity, NIH/3T3 (10^5^ cells/well) and 293T (1.5 × 10^5^ cells/well) cells were plated in 24-well plates in growth medium and incubated in a 5% CO_2_ incubator at 37 °C for 24 h. The next day, when the cells reached 40–60% confluence, the transfection complexes containing H2A or H2A-YG2 in various protein to pDNA N/P charge ratios (2.5, 5, 10, and 15) were prepared in PBS (1 μg of pDNA/well; final volume of 125 μL for two wells) and incubated at 25 °C for 30 min. Transfection with Lipofectamine^®^ 2000 was used as a positive control, while transfection with pDNA alone was used as a negative control, and all transfections were performed in duplicate. After incubation, the volume of transfection mixes was adjusted to 365 μL with serum-free medium (Opti-MEM), and the growth medium in the wells was changed to 180 μL of transfection mix. The cells were incubated with the transfection mix for 3 h in the 5% CO_2_ incubator at 37 °C, and then 0.5 mL of complete medium was added to each well. After transfection, the cells were incubated in the 5% CO_2_ incubator at 37 °C for 24 h, the medium in the wells was changed to fresh medium, and cells were further incubated for 24 h. After that, cells were washed with 0.5 mL of PBS and then lysed using 100 μL of 1× PLB/well. Prior to the measurement of luciferase activity, the lysates were centrifuged to pellet, and the insoluble debris was removed. The transfection experiments were performed in triplicate to quantify the luciferase activity.

To estimate EGFP expression using fluorescence microscopy and flow cytometry, transfections with histones in N/P 10, Lipofectamine^®^ 2000, and pDNA alone were performed, as described above. Two days after the transfection, cells were imaged using a ZOE™ Fluorescent Cell Imager (Bio-Rad) in a bright field and GFP-fluorescence channel. The images obtained from ZOE^TM^ were then processed and combined. After imaging, the cells were washed with 0.5 mL of PBS, trypsinized using 150 μL of Trypsin-EDTA, and resuspended in 350 μL of FACS buffer (0.1% FBS and 2 mM EDTA in PBS) for flow cytometric analysis. 

Plasmid DNA internalization was also analyzed using flow cytometry. Toward this end, pCMV-EGFP-P2A-luc2 was labeled with tetramethyl(TM)-rhodamine using *Label* IT^®^ Nucleic Acid Labeling kit (Mirus Bio LLC, Madison, WI, USA). NIH/3T3 (1.5 × 10^5^/well) and 293T (2 × 10^5^/well) cells were plated in 24-well plates in growth medium and incubated in the 5% CO_2_ incubator at 37 °C for 24 h. The next day, when the cells reached 60–80% confluence, transfections with histones in N/P 10, Lipofectamine^®^ 2000, pDNA alone, and Opti-MEM (serum-free medium) were performed using TM-rhodamine-labeled pDNA. All the transfections were performed in duplicate. The cells were incubated with the transfection mix for 3 h in the 5% CO_2_ incubator at 37 °C or for 1 h at 4 °C. After transfection, the plates were placed on ice, and the cells were washed with 0.5 mL of ice-cold heparin solution (1 mg/mL in PBS), followed by a wash in 0.5 mL of ice-cold PBS. Then, the cells were trypsinized using 150 μL of Trypsin-EDTA, resuspended in 350 μL of FACS buffer (0.1% FBS and 2 mM EDTA in PBS), and precipitated by centrifugation at 1000 rpm for 5 min. The cell pellet was washed with ice-cold PBS and resuspended in 0.5 mL of FACS buffer for flow cytometric analysis.

### 2.9. Luciferase Activity Assay

The transfection efficiency was analyzed using a Dual-Luciferase Reporter Assay System (Promega) according to the manufacturer’s protocol with a GENios Pro luminometer (Tecan, Mannedorf, Switzerland). Luciferase activity was measured in 10 μL of the cell lysates and normalized to the total amount of protein in the lysate. The amount of protein was quantified using the Protein Assay Dye Reagent Concentrate (Bio-Rad) according to the manufacturer’s protocol.

### 2.10. Flow Cytometry

To evaluate the presence of PDGFRβ on the surface of the cells, CT 26 and NIH/3T3 cell lines were stained with anti-PDGFRβ antibodies conjugated with phycoerythrin (PE) (ab93534, Abcam, Cambridge, MA, USA) and then analyzed by flow cytometry. The cells were washed with PBS, detached with PBS containing 2 mM EDTA, washed with PBS, and stained with antibodies at a 1:150 dilution in FACS buffer (0.1% FBS and 2 mM EDTA in PBS) for 30 min on ice. The cells were then washed with FACS buffer three times and used for flow cytometric analysis. Cell suspensions were analyzed using BD FACSAria™ III (BD Biosciences, Franklin Lakes, NJ, USA). Twenty thousand events were collected for each sample. The acquired data were analyzed using the Flowing Software 2.5.1. (Mr. Perttu Terho, Turku Centre for Biotechnology, Turku, Finland). The debris and dead cells were excluded based on FSC (forward scatter) and SSC (side scatter) coordinates. Histograms for unstained and stained cells in PE (λ_excitation_ = 561 nm, detection at 582/15 nm) channel were acquired.

To analyze the EGFP expression, the cell suspensions obtained after transfections were analyzed using FACScan^TM^ (Becton Dickinson Biosciences, San Jose, CA, USA). Ten thousand events were collected for each sample. The acquired data were analyzed using the Flowing Software 2.5.1. The debris and dead cells were excluded based on FSC and SSC coordinates. Next, based on the dot-plot in FL1 (detection at 530/30 nm)-FL2 (detection at 585/42 nm) coordinates for pDNA-transfected cells, EGFP-positive gate was established (λ_excitation_ = 488 nm). The proportion of EGFP-positive cells and overall EGFP fluorescence (proportion multiplied by mean fluorescence intensity) were analyzed for each transfection point.

Cell suspensions obtained for internalization studies were analyzed using BD FACSAria™ III (BD Biosciences). Twenty thousand events were collected for each sample. The acquired data were analyzed using the Flowing Software 2.5.1. The debris and dead cells were excluded based on FSC and SSC coordinates. Next, based on dot-plot in TM-rhodamine (λ_excitation_ = 561 nm, detection at 582/15 nm) and FSC coordinates for Opti-MEM, TM-rhodamine-positive gate was established. The proportion of TM-rhodamine-positive cells and overall TM-rhodamine fluorescence (proportion multiplied by mean fluorescence intensity) were analyzed for each transfection point.

### 2.11. Statistical Analysis

To estimate the difference between the different pairs of samples, Student’s *t*-test (two-sample assuming unequal variances) was used where applicable. Differences were considered statistically significant at *p* (one-tail) < 0.05.

## 3. Results

### 3.1. Cloning, Expression, and Purification of H2A-YG2

The DNA fragment encoding the H2A-YG2 protein was cloned into pET30a(+) using the cloning strategy shown in Figure 1a. The amino acid sequence of H2A-YG2 is shown in Figure 1b. H2A-YG2 was expressed in *E. coli* and purified using ion-exchange chromatography (IEC), followed by reversed-phase HPLC (RP-HPLC). The target protein-containing fractions from IEC and RP-HPLC are shown in Figure 1c. The protein yield was about 9.6 mg/L. H2A was expressed and purified the same way. The yield of the purified histone H2A was about 12.7 mg/L. The purity of the final preparations of H2A-YG2 and H2A was estimated using SDS-PAGE and was above 90% (Figure 1c). The yield and purity of histone H2A corresponded well with that obtained using the previously described histone purification method [5]. The molecular weight of H2A-YG2 was found to be approximately 16 kDa using SDS-PAGE, which is close to the theoretical value of 15,973 Da.

### 3.2. DNA Neutralization by Histones H2A and H2A-YG2

The negative charge of DNA prevents its free entry through the cell membrane into cells. Therefore, charge neutralization of DNA is a crucial factor for its transfer into cells. The ability of the obtained histones to condense pDNA and neutralize its negative charges was evaluated using agarose gel mobility assay. Various amounts of proteins (H2A or H2A-YG2) were mixed with 1 µg of pDNA to form complexes at different N/P charge ratios (0.5, 1, 2, 3, and 4). The results of histone/pDNA complexation revealed that the movement of the DNA was retarded completely at N/P ratios ≥ 2 (Figure 2). 

### 3.3. Characterization of the Physicochemical Parameters of the Complexes of Histones H2A and H2A-YG2 with Plasmid DNA 

We also investigated the physicochemical parameters (particle size, polydispersity index (PDI), and zeta potential) of the complexes of histones with pDNA in various N/P ratios (2.5, 5, 10, and 15). In this experiment, pDNA pCMV_EGFP_P2A_luc2 encoding a bicistronic reporter: EGFP and firefly Luc2 luciferase linked by porcine teschovirus 2A peptide was used. Most of the complexes (except for H2A and H2A-YG2 with N/P = 2.5) showed bimodal size distribution; therefore, the mean size of the particles in both peaks in corresponding cases is shown in Table 2, as well as PDI and zeta-potential of all the complexes. The percentage of the particles involved in each of the two peaks is also shown in Table 2. 

We can see that different complexes were formed at different N/P ratios. The PDI was minimal for H2A and H2A-YG2 with N/P 2.5, which was in agreement with the unimodal particle size distribution of these complexes. The size of the complexes was between 72.1 and 701.3 nm for histone H2A and between 93.1 and 760.5 nm for histone H2A-YG2 within an investigated range of N/P ratio. We did not find any correlation between the complex hydrodynamic diameters and N/P ratio or histone type. All the complexes had positive zeta-potential, which might indicate their positive surface charge that could promote the entry of the complexes into cells. The zeta-potential of the H2A-YG2:pDNA complexes with N/P 10 and 15 was comparable with that of H2A:pDNA complexes at the same N/P. The ability of H2A:pDNA complexes with N/P 10 and 15 to transfect eukaryotic cells has been previously shown [5]. Therefore, both histones formed positively-charged nano-sized complexes with pDNA that could be applicable for cell transfection. 

### 3.4. PDGFRB Expression Profiling of Different Cell Lines

To identify whether the gene delivery with the modified histone is PDGFRβ-mediated, we used cell lines differing in their PDGFRβ expression. Based on databases EBI [17] and FANTOM5 [18] and recently published data, human melanoma A-375 [19] and human embryonic kidney 293T [20] cell lines were preliminarily considered to be PDGFRβ-negative, while murine colon carcinoma CT26 [21] and murine embryonic fibroblasts NIH/3T3 [22] cell lines were assumed to be PDGFRβ-positive. The expression of the receptor RNA was examined in all four candidate cell lines using RT-PCR (Figure 3a,b). In addition, the expression of the receptor on the surface of murine cell lines was examined by staining with anti-PDGFRβ antibodies, followed by flow cytometry (Figure 3c). The maximum receptor mRNA level and surface protein staining were observed in NIH/3T3 cells, while minimal *PDGFRB* mRNA level was observed in 293T cells. Therefore, we chose NIH/3T3 as a PDGFRβ-positive cell line and 293T as a PDGFRβ-negative cell line. It is worth noting that NIH/3T3 cells are also fibroblasts and, therefore, phenotypically resemble the CAFs.

### 3.5. Transfection Studies in NIH/3T3 and 293T Cell Lines

The chosen model cell lines were transiently transfected with pDNA:histone complexes at various N/P ratios (2.5, 5, 10, and 15). PDNA pCMV_EGFP_P2A_luc2 encoding a bicistronic reporter: EGFP and firefly Luc2 luciferase linked by porcine teschovirus 2A peptide was used. The luciferase expression was evaluated using luciferase activity assay (Figure 4).

Naked pDNA did not get transfected into the cells of either cell line: the normalized luciferase activity reached 411.85 ± 207.34 RLU/mg in the case of NIH/3T3 and 0.19 ± 0.17 RLU*10^6^/mg in case of 293T. In both cell lines, when transfected with H2A-YG2:pDNA complexes, the highest level of luciferase activity was observed at the N/P ratio of 10. The effectiveness of histonefection was different in the model cell lines: luciferase activity increased up to 5-fold due to histone modification with YG2 in PDGFRβ-positive cells, while in the case of PDGFRβ-negative cells, significant changes were not detected. It can be assumed that the observed difference between histones is associated with different levels of PDGFRβ expression in these cells. Luciferase activity in the lysates of the cells transfected with Lipofectamine^®^ 2000 was significantly higher than that with histones in the case of the NIH/3T3 cell line (*p* < 0.01 for all transfection points) and was 281.80 ± 130.66 RLU*10^6^/mg, while in the case of 293T, a significant difference was not detected, and the luciferase activity was 3705.00 ± 2241.52 RLU*10^6^/mg. Similar results were observed earlier by our group for a wide range of cell lines [5].

To confirm the obtained data, the expression of EGFP in cells transfected with histone:pDNA complexes at N/P=10 was studied by fluorescent microscopy (Figure 5a). The proportion of transfected cells (Figure 5b) and overall EGFP fluorescence (Figure 5c) were estimated by flow cytometry. According to the data acquired by both fluorescence microscopy (Figure 5a) and flow cytometry (% of cells = 0 for NIH/3T3 and 0.1 for 293T; overall EGFP fluorescence = 0 a.u. for NIH/3T3 and 3 a.u. for 293T), naked pDNA was not visibly transfected into either NIH/3T3 or 293T cell lines. Fluorescence microscopy revealed a visible increase in the number of EGFP-positive cells in the case of NIH/3T3 cells following histone modification with YG2, while in the case of 293T, the increase was not noticeable. These data were then confirmed by flow cytometry. The number of transfected cells increased 3.1-fold due to histone modification in the case of NIH/3T3 cells (*p* < 0.05), while in the case of 293T, the observed increase was not statistically significant. The overall fluorescence of EGFP, in turn, increased 7-fold in PDGFRβ-positive cells (*p* < 0.05), whereas in the case of PDGFRβ-negative cells, the observed increase was not statistically significant. Thus, modification of H2A with YG2 significantly increased the proportion of transfected cells, overall EGFP production, and luciferase activity in PDGFRβ-positive cells. The observed results suggested that the complexes containing the modified histone could bind to PDGFRβ on the cell surface, which led to improved internalization of the complexes and resulted in a higher level of reporter protein production. 

We compared the results obtained with histonefection with those obtained with Lipofectamine^®^ 2000 and found that the latter transferred DNA into cells much more efficiently than histones. The proportion of cells transfected with Lipofectamine^®^ 2000 was 64.4 ± 1.7% in the case of NIH/3T3 and 99.1 ± 0.1% in the case of 293T cells. The overall fluorescence of cells transfected with Lipofectamine^®^ 2000 was 135,246 ± 1789 a.u. in the case of NIH/3T3 and 444,698 ± 578 a.u. in the case of 293T. The results for NIH/3T3 corresponded well with the results observed using luciferase activity assay, that is, the Lipofectamine^®^ 2000 transfection of cells was noticeably more effective than that with the histones. At the same time, according to flow cytometry, the efficiency of histonefection of 293T cells was significantly lower than the efficiency of lipofection (*p* < 0.01 for both proportion and overall fluorescence), whereas, according to the luciferase activity assay, no significant difference between histonefection and lipofection was detected.

### 3.6. Cellular Uptake of Histone Complexes with pDNA

To better understand the mechanism underlying the enhancement in total reporter protein production, the cellular uptake of histone:pDNA complexes was analyzed. Histone complexes with TM-rhodamine-labeled pDNA were incubated with the model cell lines, and the cellular uptake of the DNA was evaluated using microscopy (Figure 6a) and flow cytometry (Figure 6b–d). Following 3 h of incubation of the cells with the complexes, labeled DNA was detected in the cells as part of the vesicle-like structures and not distributed throughout the cytoplasm (Figure 6a). Accordingly, quantitative analysis using flow cytometry showed that overall H2A-YG2:pDNA uptake by the PDGFRβ-positive cells was about 50% (*p* < 0.05) higher than that of H2A:pDNA (Figure 6b). However, histone modification slightly reduced the number of cells that internalized the complexes (Figure 6c). Histone modification did not affect histone:pDNA uptake by the PDGFRβ-negative cells (Figure 6d,e). Further, the uptake of H2A:pDNA and H2A-YG2:pDNA complexes was not observed in either cell line at a low temperature (+4 °C), which suggests that the histone complexes enter the cells through energy-dependent endocytosis. This may also indicate that the effect of YG2 on the internalization also depends on endocytosis. These results support our hypothesis that the complexes containing the modified histone can bind to PDGFRβ on the cell surface, thus enhancing the internalization of the complexes due to endocytosis and improving transfection efficiency. 

Interestingly, the uptake of the complexes with Lipofectamine^®^ 2000 did not differ from that of histone complexes (Figure 6b–d). The internalization of Lipofectamine:pDNA complexes was significantly decreased at low temperature, and the fluoro-labeled DNA showed the same pattern of distribution within cells as in the case of histonfections (in vesicle-like structures). These results suggest that Lipofectamine:pDNA complexes internalize into cells by endocytosis. However, in the case of NIH/3T3, the luciferase activity of the cells transfected with Lipofectamine^®^ 2000 was significantly higher than that in cells transfected with histones. Moreover, the proportion of EGFP-positive cells and overall EGFP fluorescence in cells transfected with Lipofectamine^®^ 2000 were significantly higher than those in cells transfected with histones for both cell lines. Significant differences in the reporter protein expression, apparently, are not associated with the penetration of protein:DNA complexes through the cell membrane, but are caused by other factors that require further study. Differences in the transfection efficiency between histones and Lipofectamine^®^ 2000 may be explained by the different rates of endosomal escape following the internalization of the complexes.

## 4. Discussion

The efficiency of delivery into eukaryotic cells remains a major challenge when using cancer gene-therapeutic drugs in medicine [23]. Viral vectors demonstrate the high efficiency of the delivery of genetic drugs into cells, but they are characterized by low packaging capacity, relatively high production costs, immunogenicity, and toxicity that can provoke inflammation [24]. Although the first gene therapies in the market were based on viral carriers (Gendicine, Oncorine, Imlygic), the problems mentioned above seem to limit their wider application. Extensive efforts for the development of non-viral carriers for gene delivery are ongoing. Synthetic polycationic compounds, lipids, and peptides are now being increasingly used as alternative carriers. Non-viral vectors are advantageous because of their low immunogenicity (which makes them potentially safer for clinical use), practically unlimited packaging capacity, simplicity, and low-cost production [25]. It may be safer to use natural polycationic proteins as well as histones because of their natural DNA-binding function and formation of strong complexes with DNA molecules in the cell. Moreover, histones contain intrinsic NLS [26], which may facilitate the entry of therapeutic DNA-histone complexes into the nucleus. We previously demonstrated the ability of histone H2A to transfect different eukaryotic cells [5], which corresponded well with the results of Balicki and Beutler [27]. 

A common disadvantage of non-viral carriers is the low gene transfer efficiency [3]. A practical solution to this problem involves the incorporation of a domain capable of binding to the surface of cells, thus enhancing the internalization of gene-therapeutic drugs into cells [28]. We previously [5] showed that the linking of histone H2A with HIV TAT-peptide enhanced the transfection efficiency of histone, likely due to the high positive charge of the peptide, allowing efficient binding to negatively charged cell membranes and subsequent internalization of the complexes. TAT-peptide was also used to enhance the efficiency of polyethyleneimine (PEI)-polyethylene glycol (PEG) copolymer and successfully applied for DNA delivery during suicide gene therapy in a murine tumor model [29]. Ligands of cell surface receptors that induce their internalization are also frequently used as non-viral carriers. These ligands can also target DNA delivery toward certain types of cancer cells [7,30]. In this work, we constructed a non-viral carrier that targets the cells in the tumor microenvironment, more specifically, the CAFs. To achieve this, we spliced histone H2A with the peptide ligand, YG2, capable of binding to PDGFRβ, a surface marker on CAFs.

Using this approach, we generated a new recombinant protein H2A-YG2 and demonstrated that this histone H2A modification noticeably increased the transfection efficiency of PDGFRβ-positive murine fibroblasts NIH/3T3, but not that of PDGFRβ-negative human embryonic kidney 293T cells. To better understand the mechanism underlying the enhanced transfection efficiency, we studied the internalization of DNA:histone complexes in both the cell lines. The presence of the peptide increased the amount of DNA internalized by NIH/3T3 cells, but not by 293T cells. The observed internalization was energy-dependent because it was significantly inhibited at low temperatures (+4 °C). These results suggest that the H2A-YG2 in the complexes binds to PDGFRβ on the cell surface through the YG2 peptide and thereby induces receptor-mediated endocytosis of the complexes, resulting in greater transfection efficiency compared to histone H2A alone.

The CAF-targeted non-viral DNA carrier developed in this study remains to be tested in vivo. Administration of H2A-YG2:DNA complexes into a tumor, consisting of different types of cells, will provide a better assessment of the stroma specificity of the carrier. The transfection efficiency of the carrier was relatively low compared to that of Lipofectamine^®^ 2000, despite similar internalization efficiency. These results indicate that the histone complexes may have a lower endosomal escape rate. The endosomal escape could potentially be increased using endosomolytic agents, such as Ca^2+^ [31] and chloroquine [32], or through endosomolytic domains attached to the carrier [33]. The successful application of this approach has been previously demonstrated using other non-viral histone-based carriers [31,32,33].

## 5. Conclusions

This work demonstrated the fundamental possibility of targeting non-viral carriers to CAFs using PDGFRβ-binding ligands. Using the linear peptide, YG2, we observed a several-fold improvement in transfection efficiency. This peptide is characterized by a relatively high Kd (0.57 μM), although peptide ligands with better binding characteristics have been described. For example, the affibody molecules, generated by Lindborg et al. [11], had affinities of 0.4–0.5 and 6–7 nM for human and murine PDGFRβ, respectively. It can be assumed that the fusion of such an affibody with histone H2A may result in a carrier with an increased transfection efficiency of PDGFRβ-positive cells. The main part of the carrier could also be altered. For example, synthetic carriers, such as PEI-PEG copolymers, could be modified with PDGFRβ-binding ligands. Such carriers may be more effective than histone carriers. We believe that the results described in this study will prove useful to researchers working on the development of effective non-viral delivery vehicles for gene therapy. 

## Figures and Tables

**Figure 1 polymers-12-01695-f001:**
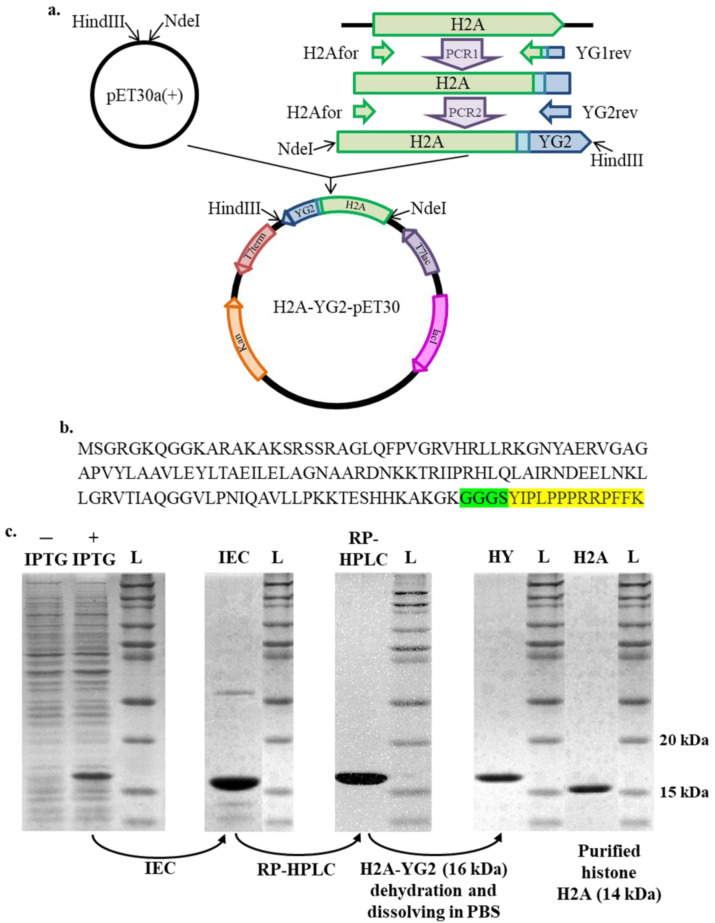
Cloning and production of recombinant histone H2A fused to YG2. (**a**)—H2A-YG2 cloning strategy, (**b**)—An amino acid sequence of H2A-YG2, histone H2A sequence is not highlighted, the linker sequence is highlighted in green, and the YG2 sequence is highlighted in yellow. (**c**)—Purification of H2A-YG2 from the clarified lysate. Lysates of *E. coli* cells without (-IPTG) and with (+IPTG) addition of IPTG, H2A-YG2 containing fractions from ion-exchange chromatography (IEC), reverse-phase HPLC (RP-HPLC), and final protein preparation (HY, 16 kDa) are shown. Purified histone H2A (H2A, 14 kDa) is also shown. The Novex Sharp Pre-Stained Protein Standard (L) is shown as a ladder.

**Figure 2 polymers-12-01695-f002:**
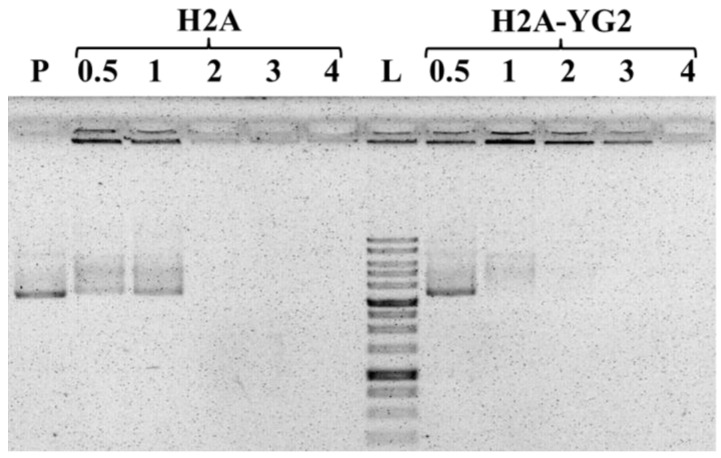
DNA neutralization by histones at different histone-DNA ratios studied by agarose gel electrophoresis. A total of 200 ng of plasmid DNA (P), SibEnzyme 1 Kb DNA Ladder (L), and complexes of histones H2A and H2A-YG2 at various N/P charge ratios (0.5, 1, 2, 3, and 4) with plasmid DNA are shown.

**Figure 3 polymers-12-01695-f003:**
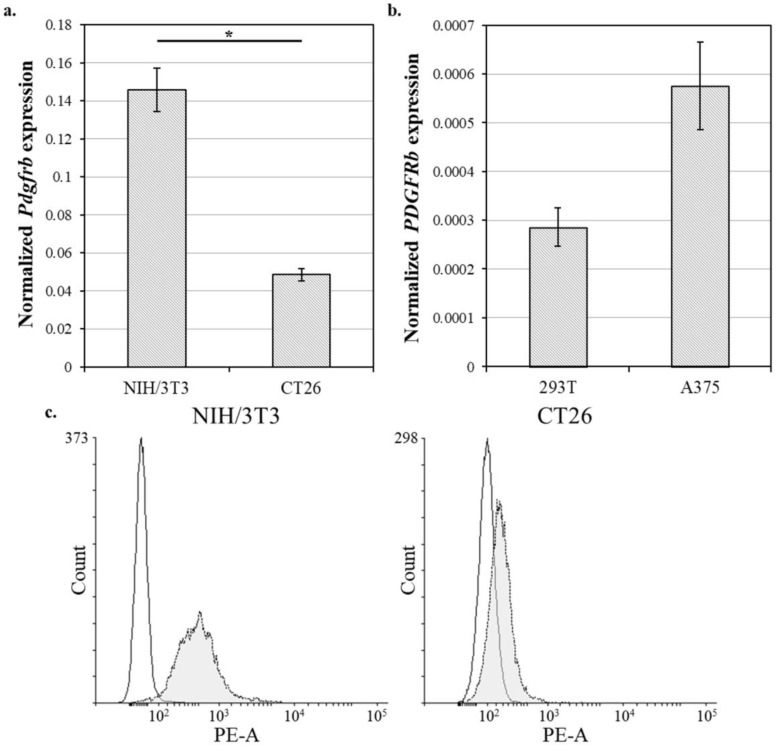
*PDGFRB* (β-type platelet-derived growth factor receptor) expression profiling of different cell lines. (**a**)—Relative mRNA level of *Pdgfrb* in murine cell lines, (**b**)—Relative mRNA level of *PDGFRB* in human cell lines. Mean values (n = 3) are shown, error bars represent standard error (n = 3); * *p* < 0.05. (**c**)—Flow cytometric analysis of NIH/3T3 and CT26 cell lines, cells without staining (open histogram) and stained with anti-mouse CD140b-(PDGFRB)-PE antibodies (filled histogram) are shown.

**Figure 4 polymers-12-01695-f004:**
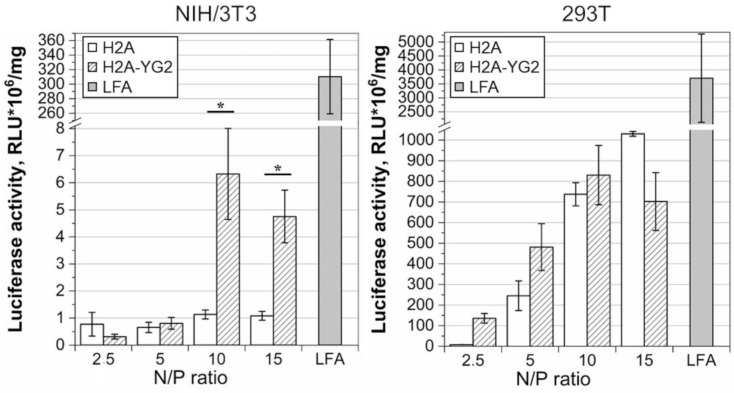
Luciferase activity in the transfected cell lines. Luciferase activity in the lysates of NIH/3T3 (Pdgfrβ-positive) and HEK 293T (PDGFRβ-negative) cells after transfection with Lipofectamine^®^ 2000 (LFA) and histones H2A and H2A-YG2 in N/P ratios 2.5, 5, 10, and 15 using pCMV_EGFP_P2A_luc2 is shown. Mean values (n = 6) are shown, error bars represent standard error (n = 6); * *p* < 0.05.

**Figure 5 polymers-12-01695-f005:**
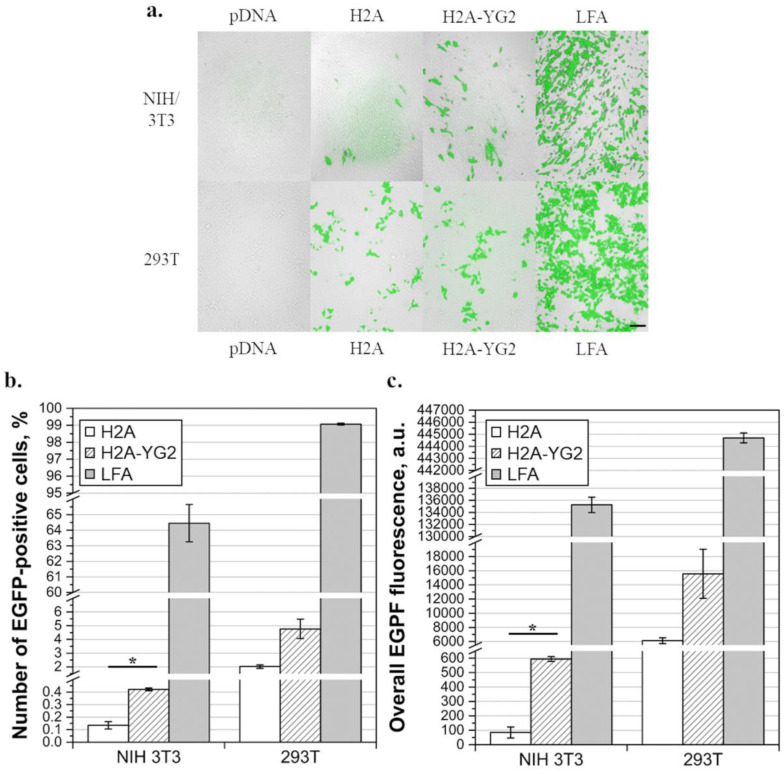
EGFP (enhanced green fluorescent protein) production in transfected cells. (**a**)—Fluorescence microscopic images of transfected cells (combined microphotographs obtained using bright field and fluorescence microscopy in the GFP-fluorescence channel). Images of NIH/3T3 and 293T cells, following transfection with pCMV_EGFP_P2A_luc2 alone (pDNA), complexed with histones H2A (H2A) or H2A-YG2 (H2A-YG2) in N/P = 10, or Lipofectamine^®^ 2000 (LFA), are shown. Scale bar equals 100 µm and is applicable to all images. (**b**,**c**)—EGFP production analyzed using flow cytometry; the proportion of EGFP-positive cells (**b**) and overall EGFP fluorescence (**c**) in NIH/3T3 and 293T cells transfected with LFA and histones H2A and H2A-YG2 in N/P = 10, using pCMV_EGFP_P2A_luc2, are shown. Mean values (n = 2) are shown, error bars represent standard error (n = 2); * *p* < 0.05.

**Figure 6 polymers-12-01695-f006:**
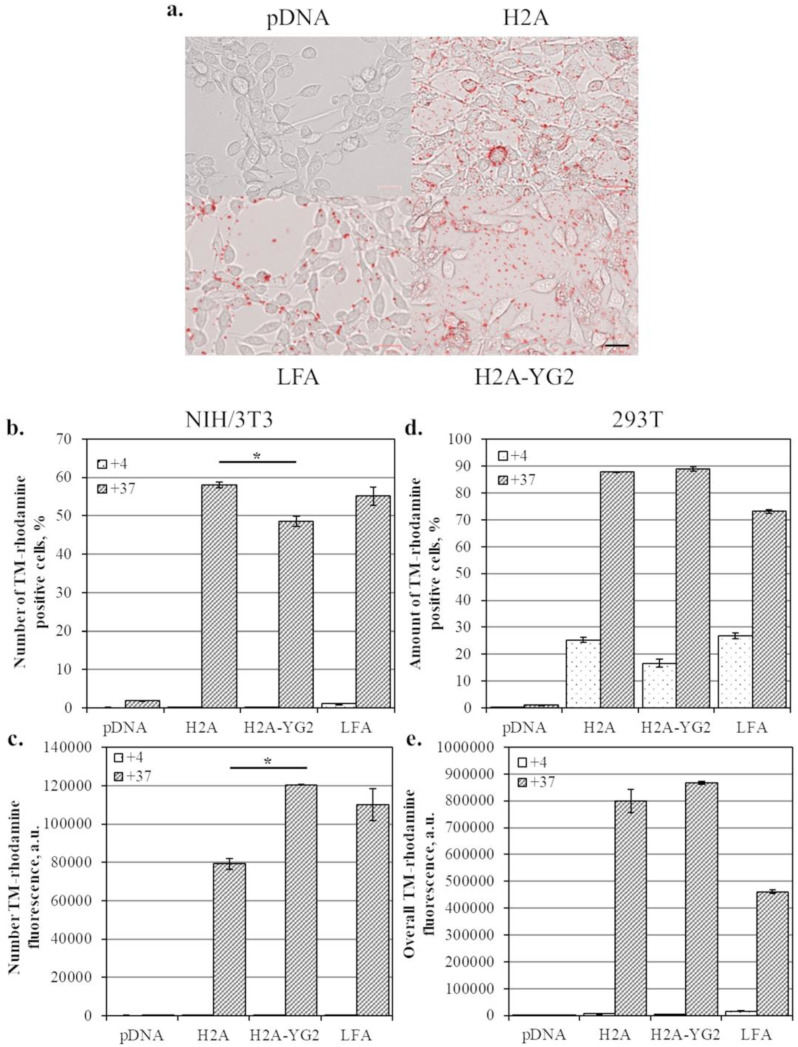
Quantitation of TMrhodamine-labeled plasmid internalization. (**a**)—Fluorescence microscopic images of NIH/3T3 cells transfected with TM-rhodamine-labeled pDNA. Images in the figure represent the combined microphotographs obtained using bright field and fluorescence microscopy in the TM-rhodamine fluorescence channel. Images of NIH/3T3 cells after transfection with pCMV_EGFP_P2A_luc2 alone (pDNA), complexed with histones H2A (H2A) or H2A-YG2 (H2A-YG2) in N/P = 10, or Lipofectamine^®^ 2000 (LFA) are shown. Scale bar equals 25 µm and is applicable to all images. The proportion of TM-rhodamine-positive cells (**b**,**d**) and overall EGFP fluorescence (**c**,**e**) in NIH/3T3 (**b**,**c**) and 293T (**d**,**e**) cells transfected with histones H2A and H2A-YG2 at N/P = 10, Lipofectamine^®^ 2000, or plasmid DNA alone using TM-rhodamine-labeled pCMV_EGFP_P2A_luc2 at +4 °C (+4) and +37 °C (+37) are shown. Mean values (n = 2) are shown, error bars represent standard error (n = 2); * *p* < 0.05.

**Table 1 polymers-12-01695-t001:** Primer sequences.

Primer Name	Primer Sequence (5′→3′)
H2A for	ATA CAT ATG TCG GGA CGT GG
YG1 rev	GAC GTG GCG GTG GCA ACG GGA TAT AGC TAC CAC CAC CCT TGC CCT TGG CCT TGT GGT
YG2 rev	AGA AAG CTT CTA CTT GAA GAA TGG GCG ACG TGG CGG TGG CAA CG
Psmb2 for	CCC AGA CTA TGT CCT CGT C
Psmb2 rev	TAC AGT GTC TCC AGC CTC TC
Psmb7 for	GCG GCT GTG TCG GTG TTT C
Psmb7 rev	CCT TCA GTT GCT CTC GTG TC
Pdgfrb for	CTG CTG GAG ACA CTG GGA G
Pdgfrb rev	CGC TTC TGA CAC CTT CAC AC
HPRT for	GCT ATA AAT TCT TTG CTG ACC TGC TG
HPRT rev	AAT TAC TTT TAT GTC CCC TGT TGA CTG G
PDGFRB for	CGT CAA GAT GCT TAA ATC CAC AG
PDGFRB rev	GAT GAT ATA GAT GGG TCC TCC TTT G
18S for	TGC AAT TAT TCC CCA TGA ACG
18S rev	GCC TCA CTA AAC CAT CCA ATC

**Table 2 polymers-12-01695-t002:** Physicochemical parameters of the histone:pDNA complexes. Mean hydrodynamic diameter of particles in the observed peaks (d1 and d2), % of particles in the peaks, polydispersity index (PDI), and zeta-potential of the complexes at different N/P charge ratios (2.5, 5, 10, 15) are shown. Mean values are shown; where applicable, the result is shown as mean ± standard deviation.

Protein	N/P Ratio	d1, nm	% in Peak 1	d2, nm	% in Peak 2	PDI	Zeta-Potential, mV
H2A	2.5	292.8 ± 1.1	100.0	−	−	0.005	8.36 ± 1.87
	5	93.5 ± 7.1	94.9	311.3 ± 22.0	5.1	0.122	26.21 ± 9.73
	10	72.1 ± 6.0	97.1	281.6 ± 17.8	2.9	0.154	27.0 ± 13.65
	15	223.9 ± 15.3	93.1	701.3 ± 56.2	6.9	0.164	35.16 ± 20.26
H2A-YG2	2.5	774.8 ± 6.2	100.0	−	−	0.082	26.30 ± 3.72
	5	147.8 ± 9.7	78.9	760.5 ± 86.3	21.1	0.178	20.62 ± 6.56
	10	181.7 ± 8.8	31.1	440.4 ± 17.6	68.9	0.122	25.28 ± 6.18
	15	93.1 ± 5.1	86.5	273.6 ± 17.2	13.5	0.125	31.87 ± ND ^1^

^1^ These complexes were susceptible to the electric field used for zeta-potential measurement; therefore, the zeta-potential value for this sample is approximate.

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
