# Peer review of "Novel Histone-Based DNA Carrier Targeting Cancer-Associated Fibroblasts"

_polymers, 2020, doi:10.3390/polym12081695_

Round 1

Reviewer 1 Report

Sverdlov et al. reported the preparation of H2A-YG2 fusion protein by the means of prokaryotic expression. The as-prepared H2A-YG2 could bind pDNA and mediate targeted gene delivery to eukaryotic cells by the specific interactions between YG2 and beta-type platelet-derived growth factor receptors on cells. The paper should address the following issues before being considered for publication.

  1. What is the advantage of prokaryotic expression over chemical synthesis? Why did the authors use prokaryotic expression to prepare H2A-YG2 without using chemical conjugation?
  2. The NLS signal in H2A-YG2 should also be highlighted in Figure 1b.
  3. What are the size, zeta potential and PDI of the DNA/ H2A-YG2 complexes?
  4. Why did the H2A and H2A-YG2 have higher gene transfection efficiency on 293T cells than on 3T3 cells?
  5. Please provide the TEM microscopy results of DNA/ H2A-YG2 complexes.

Reviewer 2 Report

I have attached a file with my comment.

Reviewer 3 Report

The current manuscript describes the generation and optimization of a histone-based DNA transfection method as a potential non-viral alternative for gene therapy. The authors generate a complex histone-DNA modified with a peptide targeting the platelet-derived growth factor receptor beta to increase transfection efficiency in cancer associated fibroblasts, usually expressing this receptor. The article covers a topic of enormous relevance for the field of gene therapy. The experimental design is correct and the conclusions are fully supported by the data. I suggest the following modifications to improve the presentation and comparation of the main findings:

  • Luminescence and fluorescence data obtained using Lipofectamine2000 should be also added to figures 4 and 5 b-c, respectively. Currently, this information is only described in the text and presented in the microscopy images. Eventually, a break in the y-axis may be introduced if values are highly different.
  • Figures 5a and 6a. Scale bars need to be added to the microscopy images.
  • Figure 5. Although fluorescence data presented in this figure confirm the findings obtained using luminescence (Fig 4), it must be noted that only two independent experiments (n=2) were performed. Adding one more experiment (n=3) would strengthen the conclusions raised from these experiments.
  • Line 380. The sentence needs to be reformulated. Now it reads “uptake by the PDGFRβ-positive cells was 1.5 times higher than that of H2A:pDNA”. According to the figure 6b, the uptake was only 50% higher (and not 150% as the current sentence suggests).
